# An Efficient Method for Monitoring Birds Based on Object Detection and Multi-Object Tracking Networks

**DOI:** 10.3390/ani13101713

**Published:** 2023-05-22

**Authors:** Xian Chen, Hongli Pu, Yihui He, Mengzhen Lai, Daike Zhang, Junyang Chen, Haibo Pu

**Affiliations:** 1College of Information Engineering, Sichuan Agricultural University, Ya’an 625000, China; 202005726@stu.sicau.edu.cn (X.C.); 202005752@stu.sicau.edu.cn (H.P.); 202005817@stu.sicau.edu.cn (Y.H.); 202105819@stu.sicau.edu.cn (M.L.); 202005869@stu.sicau.edu.cn (J.C.); 2Ya’an Digital Agricultural Engineering Technology Research Center, Ya’an 625000, China

**Keywords:** object detection, multi-object tracking, computer vision, YOLOv7, attention mechanism, bird conservation, bird monitoring

## Abstract

**Simple Summary:**

Knowing the species and numbers of birds in nature reserves is essential to achieving the goals of bird conservation. However, it still relies on inefficient and inaccurate manual monitoring methods, such as point counts conducted by researchers and ornithologists in the field. To address this difficulty, this paper explores the feasibility of using computer vision technology for wetland bird monitoring. To this end, we build a dataset of manually labeled wetland birds for species detection and implement taxonomic counts of ten wetland bird species using a deep neural network model with multiple improvements. This tool can improve the accuracy and efficiency of monitoring, providing more precise data for scientists, policymakers, and nature reserve managers to take targeted conservation measures in protecting endangered birds and maintaining ecological balance. The algorithm performance evaluation demonstrates that the artificial intelligence method proposed in this paper is a feasible and efficient method for bird monitoring, opening up a new perspective for bird conservation and serving as a reference for the conservation of other animals.

**Abstract:**

To protect birds, it is crucial to identify their species and determine their population across different regions. However, currently, bird monitoring methods mainly rely on manual techniques, such as point counts conducted by researchers and ornithologists in the field. This method can sometimes be inefficient, prone to errors, and have limitations, which may not always be conducive to bird conservation efforts. In this paper, we propose an efficient method for wetland bird monitoring based on object detection and multi-object tracking networks. First, we construct a manually annotated dataset for bird species detection, annotating the entire body and head of each bird separately, comprising 3737 bird images. We also built a new dataset containing 11,139 complete, individual bird images for the multi-object tracking task. Second, we perform comparative experiments using a state-of-the-art batch of object detection networks, and the results demonstrated that the YOLOv7 network, trained with a dataset labeling the entire body of the bird, was the most effective method. To enhance YOLOv7 performance, we added three GAM modules on the head side of the YOLOv7 to minimize information diffusion and amplify global interaction representations and utilized Alpha-IoU loss to achieve more accurate bounding box regression. The experimental results revealed that the improved method offers greater accuracy, with mAP@0.5 improving to 0.951 and mAP@0.5:0.95 improving to 0.815. Then, we send the detection information to DeepSORT for bird tracking and classification counting. Finally, we use the area counting method to count according to the species of birds to obtain information about flock distribution. The method described in this paper effectively addresses the monitoring challenges in bird conservation.

## 1. Introduction

The global ecological environment is under severe threat due to rapid social and economic development [1,2], leading to the endangerment of many bird species [3]. Consequently, the protection and conservation of endangered organisms have emerged as one of humanity’s most pressing concerns. Numerous countries worldwide have implemented various measures to help protect birds, ensuring their reproduction and survival. Typically, monitoring stations are established in protected areas for wildlife monitoring and management. The data on bird species and populations collected through monitoring allow reserve managers to better understand bird survival and distribution patterns, enabling the development of the most effective measures for bird protection. Therefore, efficient monitoring of birds is one of the keys to addressing bird conservation issues.

Nevertheless, currently, bird monitoring methods mainly rely on manual techniques, such as point counts conducted by researchers and ornithologists in the field [4,5,6,7,8], utilizing equipment such as binoculars, high-powered cameras, and telephoto lenses to conduct fixed-point observations in areas where birds congregate. This approach is not only time-consuming and labor-intensive, but it also suffers from limited coverage, and the data obtained are often untimely, inaccurate, and incomplete. This is particularly true for near-threatened and endangered species, which may not be observed due to their low occurrence and population numbers [9]. This outdated method significantly impedes bird conservation efforts. As a result, there is an urgent need to develop efficient methods to enhance bird monitoring efficiency.

In recent years, bird monitoring technology has experienced significant advancements. For instance, Zheng Fa et al., conducted a sample line survey at field sample sites using a telephoto lens SLR digital camera and monocular and binoculars [10]. Sun Ruolei et al., employed bird songs, photographs, and professional resources, such as the “Field Manual of Chinese Birds” for identification [11]. Liu Jian et al., utilized biological foot ring sensors [12]. In addition, some scholars have also employed methods such as aerial photography, unmanned aerial vehicle (UAV) surveys [13,14,15,16,17,18,19], and the use of bioacoustics for bird monitoring [20,21,22,23]. Compared with traditional manual monitoring methods, these methods can improve the efficiency of bird monitoring and avoid unnecessary time waste. However, they rely heavily on manual labor and experience accumulation to manage bird monitoring, which is vulnerable to factors such as small bird objects, high shading, high density, harsh field environments, and heavy manual workload. Despite these improvements, the monitoring efficiency is still not high enough. Therefore, there is an urgent need for intelligent and modern methods to promote bird conservation in the direction of precision and automation.

The development of artificial intelligence has expanded from a single application area to a wide range of applications [24,25], and computer vision technology is one of them. The application of computer vision to animal protection is one of the hot spots of research among scholars all over the world [26,27]. Juha Niemi et al., investigated bird identification through a bird radar system combined with an object-tracking algorithm [28]. They applied convolutional neural networks trained by deep learning algorithms to image classification, demonstrating the need for an automatic bird identification system suitable for real-world applications. In a breakthrough in this research area, scientists from research teams at CNRS, the University of Montpellier, and the University of Porto, Portugal, have developed the first artificial intelligence model capable of identifying individual birds [29], and their system is capable of automatically identifying individual animals with no external markers at all, without human intervention, and without harming the animals. However, the system has some limitations; it can only identify birds in the database and cannot cope with changes in appearance, such as feather changes. Lyu Xiuli and colleagues from Northeast Petroleum University utilized a convolutional neural network to identify and locate red-crowned cranes [30] and established a recognition model specifically for this species that showed good identification performance for red-crowned crane populations. However, they did not develop a multi-classification and comprehensively systematic method, which can only identify red-crowned cranes and is largely impractical.

Reviewing the related works in recent years, it has been found that although many effective works have been carried out in the field of bird monitoring and protection, they still have some shortcomings, such as the fact that the practicality needs to be improved, the models are not stable enough, and the accuracy is low. In addition, there are many species of birds, which are numerous and difficult to identify. No algorithm can accurately classify birds and record their numbers in precise species. Therefore, it is necessary to study a highly efficient method with good classification performance and species counting capability to monitor birds, deepen the research on the intelligence and automation of bird conservation, and promote the conservation of biodiversity.

This study selects ten priority-protected bird species, including the Ruddy Shelduck (*Tadorna ferruginea*), Whooper Swan (*Cygnus cygnus*), Red-crowned Crane (*Grus japonensis*), Black Stork (*Ciconia nigra*), Little Grebe (*Tachybaptus ruficollis*), Mallard (*Anas platyrhynchos*), Pheasant-tailed Jacana (*Hydrophasianus chirurgus*), Demoiselle Crane (*Anthropoides virgo*), Mandarin Duck (*Aix galericulata*), and the Scaly-sided Merganser (*Mergus squamatus*), as research objects. These species are under protection due to their declining population numbers and are of great conservation concern. Monitoring bird populations is a crucial tool for protecting bird species. To better monitor these protected bird species, we propose an efficient and automated bird monitoring method based on the latest object detection and multi-object tracking technologies, which is capable of achieving precise monitoring for these ten bird species and offering a new perspective on bird monitoring. Firstly, we detect and locate the birds by object detection and obtain the species information of the birds; then, we use the multi-object tracking algorithm to assign a unique ID to each object to ensure the accuracy of counting and avoid duplicate counting or missed counting due to occlusion; and finally, we combine the detection results with the ID information to realize the classification counting of the birds. Since the performance of the tracking by the detection method depends on the quality of the object detection algorithm, we also target improving the object detection algorithm, aiming to improve the efficiency of bird monitoring, promote the research of intelligent and automated bird conservation, and protect biodiversity.

Specifically, the contribution of this paper includes the following points. Firstly, we propose a new method of bird monitoring based on object detection and multi-object tracking networks. The method improves the efficiency of bird conservation, and at the same time, it is highly portable and provides a reference for the conservation of other animals. Secondly, we improve the object detection part in this paper. In this paper, the YOLOv7 algorithm is used as the baseline for object detection, and three GAM modules are added to the head side of YOLOV7 to reduce information dispersion, amplify the global interaction representation, and replace the loss function with Alpha-IoU loss to obtain more accurate bounding box regression and object detection. In this regard, the performance of the YOLOv7 algorithm and the performance of the proposed method for bird monitoring in this paper are optimized. Thirdly, an ingenious method of sorting and counting is designed. We make a counting board of the same size as the original image and combine the detection result and the tracking-assigned ID information to realize the counting of birds by species. (The specific technical idea of counting will be described in detail in Section 2.4.4 of this paper.) Finally, a manually annotated bird species detection dataset is constructed in this paper. It contains ten species of key protected birds, including 3737 images of bird flocks, and adopts pure-head annotation and whole-body annotation methods for annotation, respectively. The dataset images have both single bird activities and dense flock activities, which are inevitably disturbed by natural factors such as vegetation shadows, non-avian animals, water bodies, and litter. These datasets are derived from various real environments in wetlands, which makes the trained model more robust and well-suited for practical use, and the dataset can also be used as a reference for bird species detection studies. We also build a new dataset for the multi-object tracking task, containing 11,139 complete individual bird images and various motion patterns and shooting angles of birds, allowing the trained model to extract more effective features and be more robust, in addition to expanding the fine-grained classification dataset of birds (e.g., “CUB-200-2011”).

## 2. Materials and Methods

### 2.1. Data Acquisition

Ten types of protected birds in China were selected as the objects of the experiment. (The names of the ten types of birds are as follows: ”Ruddy Shelduck, Whooper Swan, Red-crowned Crane, Black Stork, Little Grebe, Mallard, Pheasant-tailed Jacana, Demoiselle Crane, Mandarin Duck, Scaly-sided Merganser”.)

The dataset for this paper is divided into two sections: (1) the dataset for object detection and (2) the dataset for a multi-object tracking feature extraction network.

The data used in this paper for object detection come from the internet. In order to ensure the authenticity and validity of the experiment, we screened the quality of the data to meet the minimum pixel requirements of more than 1080P, and all of them are authentic bird images in a wetland environment. The collected data contain a variety of interference factors, such as overlap, distance change, light and shade change, vegetation shadow, non-avian animals, and garbage. These interferences can replicate many conditions in real scenes, enhance the robustness of the algorithm, and improve the generalization ability of the model to ensure the effectiveness of the method. In addition, considering the occlusion between birds, this paper not only marks the whole body of the bird but also marks the head of the bird separately.

The dataset used in the multi-target tracking feature extraction network in this paper comes from the dataset of object detection. We extract the complete image of each bird from the dataset of object detection, including multiple angles and actions of birds. The dataset can also be used to expand the fine-grained classification dataset of birds. The preview of the above two parts of the dataset is shown in Figure 1.

All datasets in this paper are divided into a training set, a validation set, and a test set according to a ratio of 85∕10∕5, which can be accessed and downloaded online at the following link: “https://www.kaggle.com/datasets/dieselcx/birds-chenxian (accessed on 12 May 2023)”. The division of the datasets is shown in Table 1 and Table 2.

### 2.2. Data Preprocessing

#### 2.2.1. Mosaic Data Enhancement

Mosaic is a data enhancement method proposed in YOLOv4 [31]. The method focuses on randomly selecting four images and splicing them into a new image as training data after transforming them by random scaling, random cropping, and random lining up.

Mosaic data enhancement has two main advantages. (1) Expanding the dataset: In addition to enriching the background of the detection dataset, random scaling also adds many small targets, making the model more robust. (2) Reducing GPU memory: In batch normalization, the data of four images are computed simultaneously, which can reduce the dependence on batch size, and a single GPU can complete the training. The workflow for Mosaic’s data augmentation operation is shown in Figure 2.

#### 2.2.2. Mixup Data Enhancement

Mixup [32] is an algorithm for mixing classes of augmentation of images in computer vision. It is based on the principle of neighborhood risk minimization and uses linear interpolation to mix images between different classes to construct new training samples and labels, which expand the training dataset. The image processing formula for the mixup data enhancement is as follows:(1)$$\tilde{x}=\lambda x_{i}+\left(1-\lambda \right)x_{j}$$
(2)$$\tilde{y}=\lambda y_{i}+\left(1-\lambda \right)y_{j}$$

$\left(x_{i},y_{i}\right)$ and $\left(x_{j},y_{j}\right)$ are two randomly selected samples and corresponding labels from the same batch, which are randomly sampled numbers from the beta distribution. λ is a parameter that follows the distribution of β,$\lambda \in \left[0,\;1\right]$. Figure 3 shows several images after the mixup data enhancement process.

#### 2.2.3. HSV Data Enhancement

HSV is a color space created based on the intuitive properties of color. H stands for hue, S for saturation, and V for value. Hue, which means color, is measured in degrees in a range of $\left[0{}^{\circ},360{}^{\circ}\right]$. We can change its color by changing the size of the angle. Saturation indicates how close a color is to a spectral color. A color can be seen as the result of mixing a certain spectral color with white. If we adjust the spectral color so that its proportion increases, then the closer the color is to the spectral color, the higher the saturation is. The range of saturation is $\left[0,\;1\right]$. The value indicates the color’s degree of brightness. The brightness value is related to the luminosity of the luminous body. If we increase luminosity, the color will be brighter, and the range of the value is $\left[0,\;1\right]$. Figure 4 shows several images after the HSV data enhancement process.

### 2.3. Related Networks

#### 2.3.1. Object Detection: YOLOv7

YOLOv7 [33] is the work of the YOLO (You Only Look Once) series and is one of the most advanced object detection models available. YOLOv1 [34] was proposed in 2015 and was the debut of a single-stage detection algorithm, which emerged to effectively address the drawback of slow inference in two-stage detection networks while maintaining good detection accuracy. Subsequently, the authors proposed an improved YOLOv2 [35] based on YOLOv1, which used a joint training method of detection and classification, enabling the model to detect more than 9000 classes of objects. Next, YOLOv3 [36] was proposed as an improved version of the previous work. Its most significant feature is the introduction of the residual module Darknet-53 and the FPN architecture, which allows object prediction and multiscale fusion at three different scales. In addition, this version also adds new tricks, such as batch normalization and the mish activation function, to further improve the detection accuracy of the YOLO series. Based on this, YOLOv4 and YOLOv5 were introduced, which added many tricks to YOLOv3. YOLOv4 introduced modules such as CSPDarknet53, SPP [37], and PAN [38], which enhanced the perceptual field and feature representation of the network. In addition, it adopts new tricks, such as the Mosaic data enhancement trick and DropBlock [39] regularization trick, to further improve detection accuracy. YOLOv5, on the other hand, adopts a large number of design tricks such as the focus structure, improved CSP module, adaptive anchor frame calculation, and adaptive image scaling, which means the model has a qualitative leap in speed and accuracy.

Finally, YOLOv7 came out in 2022 with the network architecture shown in Figure 5. Based on its predecessor, it innovatively proposes the extended ELAN architecture, which can improve the self-learning capability of the network without destroying the original gradient path. In addition, it employs a cascade-based model scaling method so that a model of the appropriate scale can be generated for the actual task to meet the detection requirements. The introduction of these new tricks and architectures further improves the performance and effectiveness of the YOLO series networks. In this paper, we take the YOLOv7 network as a baseline and further enhance it.

#### 2.3.2. Introducing Attention Mechanism into YOLOv7: GAM

The attention mechanism [40] is a signal-processing mechanism that was discovered by some scientists in the 1990s while studying human vision. Practitioners in artificial intelligence have introduced this mechanism into some models with success. Currently, the attention mechanism has become one of the most widely used “components” in the field of deep learning, especially in the field of natural language processing. Models or structures such as BERT [41], GPT [42], Transformer [43], etc., which have received much exposure in the past two years, all use the attention mechanism. It simulates the phenomenon that humans selectively pay attention to some visible information and ignore others to rationalize the limited visual processing resources. Specifically, the information redundancy problem is mainly solved by selecting only a part of the input information or assigning different weights to different parts of the input information.

In the process of exploring the application of attention mechanisms in computer vision, many excellent works have emerged, although they also have some drawbacks. For example, SENet [44] also brings the problem of low efficiency when suppressing unimportant pixels, CBAM [45] performs channel and spatial attention operations sequentially, while BAM [46] does them in parallel, but they both ignore the channel–space interaction, thus losing the cross-dimensional information. Considering the importance of cross-dimensional interactions, TAM [47] improves efficiency by exploiting the attention weights between each pair of 3D channels, spatial width and spatial height. However, the attention operation is still applied to two dimensions at a time, instead of all three dimensions. Therefore, to amplify cross-dimensional interactions, GAM [48] proposes an attention mechanism capable of capturing important features in all three dimensions, which is able to amplify global dimensional interaction features even with reduced information dispersion. The authors use a sequential channel–space attention mechanism and redesign the CBAM submodule. The whole process is shown in Figure 6, and the definitions are stated in Equations (3) and (4). Given an input feature map $F_{1}\in R^{C\times H\times W}$, the intermediate state $F_{2}$ and the output $F_{3}$ are defined as follows: $M_{C}$ and $\;M_{S}$ are the channel attention map and the spatial attention map, respectively, and ⊗ denotes the multiplication operation by the elements.
(3)$$F_{2}=M_{C}\left(F_{1}\right)⊗F_{1}$$
(4)$$F_{3}=M_{S}\left(F_{2}\right)⊗F_{2}$$

Channel Attention Sub-module

The channel attention submodule uses a three-dimensional arrangement to retain information in three dimensions. Then, it uses a two-layer MLP (Multilayer Perceptron) to amplify the cross-dimensional channel–space dependence. (MLP is an encoder–decoder structure, the same as BAM, and its compression ratio is r); the channel attention submodule is shown in Figure 7.

2.Spatial Attention Sub-module

In the spatial attention submodule, in order to focus on spatial information, two convolutional layers are used for spatial information fusion, and the same reduction ratio r as BAM is used from the channel attention submodule. At the same time, since the maximum pooling operation reduces the use of information and has a negative impact, the pooling operation is deleted to further preserve feature mapping.

In order to increase the precision of object detection, GAM is added to the YOLOv7 network in this study. The modified network structure is shown in Figure 8. There are two primary uses for the GAM module. First, it can lessen information dispersion, allowing the network to focus more on the properties of the target object and enhance detection performance. Second, it can expand the global interactive representation, increase the sufficiency of information exchange between various components, and improve the accuracy of detection. When attention mechanisms are added to the backbone network, some of the original weights of the backbone network are destroyed, which causes errors in the prediction results of the network. In this case, we decided to keep the original network features intact while incorporating the attention mechanism into the enhanced feature network extraction.

#### 2.3.3. Introducing Alpha-IoU into YOLOv7

Object detection is used to locate the object in the image by bounding box regression. In early object detection work, IoU was used as the localization loss [49]. However, when the prediction box does not overlap with the ground truth, the IoU loss will cause the problem of gradient disappearance, resulting in a slower convergence speed and an inaccurate detector. To solve this problem, researchers have proposed several improved IoU-based loss designs, including GIoU [50], DIoU [51], and CIoU [52]. Among them, GIoU introduces a penalty term in the IoU loss to alleviate the gradient disappearance problem, while DIoU and CIoU consider the center point distance and aspect ratio between the prediction box and the ground truth in the penalty term.

Alpha-IoU [53] generalizes the existing IoU-based losses into a new series of power IoU losses, which has a power IoU term and an additional power regularization term. First, the Box-Cox transform is applied to the IoU loss, and it is generalized to power IoU loss, denoted by α, and finally extended to a more general form by adding a power regularization term. In simple terms, it is a power operation in the IoU and the penalty term expression. The calculation is shown in Equation (5).
(5)$$L_{\mathrm{IoU}}=1-IoU\Rightarrow L_{\alpha \cdot \mathrm{IoU}}=1-IoU^{\alpha }$$

The Alpha-IoU loss function can generalize the existing IoU-based losses, including GIoU and DIoU, to a new power IoU loss function to achieve more accurate bounding box regression and object detection. For example, based on GIOU and DIOU, the formula changed to the corresponding Alpha-IoU is shown in Equations (6) and (7).
(6)$$L_{\mathrm{GIoU}}=1-IoU+\frac{\left|C\setminus \left(B\cup B^{gt}\right)\right|}{\left|C\right|}\Rightarrow L_{\alpha \cdot \mathrm{GIoU}}\;\;\;\;\;\;\;\;\;\;\;\;=1-IoU^{\alpha }+{\left(\frac{\left|C\setminus \left(B\cup B^{gt}\right)\right|}{\left|C\right|}1+x\right)}^{a}$$
(7)$$L_{\mathrm{DIoU}}=1-IoU+\frac{\rho^{2}\left(b,b^{gt}\right)}{c^{2}}\Rightarrow L_{\alpha \cdot \mathrm{DIoU}}=1-IoU^{\alpha }+\frac{\rho^{2\alpha }\left(b,b^{gt}\right)}{c^{2\alpha }}$$

The YOLOv7 network is given an upgrade in this paper with the introduction of Alpha-IoU, which can outperform the IoU-based loss with clear performance benefits and give the detector more flexibility by adjusting to achieve various levels of bbox regression accuracy. Additionally, it is more resistant to our datasets and noise bboxes, strengthening the model’s resistance to complex situations.

#### 2.3.4. Multi-Object Tracking: DeepSORT

The multi-object tracking task is to detect and assign IDs to multi-objects in the video for trajectory tracking without prior knowledge of the number of objects, where each object has a different ID to enable subsequent trajectory prediction, precision finding, etc. DeepSORT [54] is the most popular algorithm for multi-object tracking tasks and an improved algorithm based on the ideas of SORT [55].

The SORT algorithm uses a simple Kalman filter [56] to process the correlation of frame-by-frame data and a Hungarian algorithm for the correlation metric, a simple algorithm that achieves good performance at high frame rates. However, as SORT ignores the surface features of the object being inspected, it is only accurate when the uncertainty in the estimation of the object’s state is low. In DeepSORT, appearance information is added, the ReID domain model is borrowed to extract appearance features, the number of ID switches is reduced, a more reliable metric is used instead of the association metric, and a CNN network is used to extract features to increase the robustness of the network to misses and obstacles.

The tracking scene of the DeepSORT algorithm is defined on an eight-dimensional state space $\left(u,v,\gamma ,h,\dot{x},\dot{y},\dot{\gamma },\dot{h}\right)$, where $\left(u,v\right)$ are the coordinates of the detection frame centroid, γ is the aspect ratio, and γ is the height of the detection frame and their respective velocities in image coordinates. Then, a homogeneous model and a linear observational model Kalman filter are used with the observed variables $\left(u,v,\gamma ,h\right)$ to forecast updates. For each trajectory k, the number of matched frames is counted from the moment of the last first match ak; the count is incremented if a match is made during Kalman prediction and reset to 0 if the trajectory is associated with a new prediction. Additionally, set a lifetime threshold of Amax, after which no match of time is considered to have left the tracking area and is removed from the track (in layman’s terms, an object that has not matched for a long time beyond Amax is considered to have left the tracking area). Since each newly detected object may become a new trajectory or if they are directly classified as a trajectory, then false detection will occur frequently. DeepSORT marks the new test result as“tentative,” which is followed by a few frames (usually three) and then “confirmed” if the next three consecutive frames match; it is confirmed as a new track. Otherwise, it is marked as ‘deleted’ and is no longer considered to be a track. Figure 9 shows the bird tracking process.

The original DeepSORT’s output is not intuitive enough to display the detected species. To improve the presentation, the source code is modified in this paper to add the display of species, making the output more intuitive. Figure 10 shows the results of the bird tracks.

### 2.4. Monitoring Methods

This paper proposes a computer vision-based bird monitoring method for detecting birds in nature reserves and counting them by analogy. By combining the information from the monitoring points, bird surveyors will have information on the distribution of bird populations and migration routes so that they can develop more effective ways to protect birds. The algorithm is divided into three steps: first, using an object detection algorithm to detect the species of birds and locate the object; then, using multi-object tracking to track the birds, each bird object is assigned a unique ID for subsequent data processing and analysis; and finally, the birds are counted by species using a counting method that combines the species, location, and ID information. To better illustrate the methodology of this paper, further details of the above will be added below.

#### 2.4.1. Different Labelling Methods

In object detection, if only the bird’s body is considered for annotation, it may lead to more overlapping parts between the annotation frame and the background, thus causing too much background noise and affecting the accuracy and reliability of object detection. A bird’s head has different structural forms, which can accurately identify birds. In complex scenes, such as those with object overlap and occlusion, the annotation of a bird’s head can help to better distinguish different birds and reduce the probability of misjudgment. Therefore, in this paper, not only the bird’s body but also the bird’s head is labeled, and the two labeling methods form a set of comparative experiments to find the optimal model.

#### 2.4.2. Obtaining the Best Algorithmic Model

The dominant multi-object tracking method is based on detection to track. Therefore, the method’s effectiveness depends on object detection. To achieve better results, we need an excellent object detection model.

First, we selected a group of current mainstream object detection networks to form a set of comparative experiments, selected the network with the best results, and continued to improve the object detection part to obtain the best model, with the following ideas for improvement. (1) Data augmentation: expanding the dataset and increasing the diversity of the data by rotating, flipping, cropping, and scaling the data to improve the robustness and generalization of the model. (2) Algorithm optimization: optimization of the object detection algorithm, e.g., improving the loss function, network structure, optimizer, etc., to improve the training efficiency and detection accuracy of the model. (3) Feature fusion: using multiple feature maps for object detection and fusing features at different levels and scales to improve the model’s ability to perceive and recognize objects.

The above measures will improve the accuracy and precision of bird monitoring, providing more accurate data to support bird research and conservation work.

#### 2.4.3. Multi-Object Tracking

Multiple detected objects are given unique identifiers, and trajectory tracking is carried out. Each object has a different ID to enable subsequent counting, accurate searching, and so on.

#### 2.4.4. Implementation of the Counting Area

To count the different species of birds separately and more accurately, a new method is proposed in this paper. The method uses an area that covers 95% of the image for counting, which ensures a more accurate count of the birds. Specifically, when a bird enters the counting area, we read information about the bird’s species and ID, which is fed back using the computer vision method described above. If the ID is a first occurrence, we record the ID and increase the number of birds in that species; if the ID has already been recorded, we leave the number of birds recorded for that species unchanged. In this way, we can more accurately count birds by species and, therefore, obtain a more accurate count.

For the logical implementation of the counting area, this paper first makes a matrix of the same size as the image and fills it with 1 to form the counting area. Next, we replace the value of the area we’re counting with a 0 and reassign the detected bird’s position information passed back to the corresponding position on our matrix to 1 to mark the presence of the bird in that area. Finally, we judge whether to count according to whether there is a 1 in the counting area. The counting method realized by this logic can effectively record the number of birds. The overall logical implementation diagram of the counting method is shown in Figure 11, which clearly shows the counting process of the method.

## 3. Results

### 3.1. Experimental Environment

Table 3 below shows the basic equipment information of the software and hardware used in this paper.

### 3.2. Training Parameters

Table 4 shows the training parameters of the training process used in the experiment.

### 3.3. Evaluation Metrics

In this paper, we use mainstream evaluation metrics such as precision, recall, F1 score, mAP, and FPS to evaluate the effect of the model. Before introducing each evaluation metric, we briefly present the confusion matrix, whose parameters are defined in Table 5.

Precision indicates the proportion of samples that the model correctly identifies as belonging to positive classes. It reflects the ability of the model to distinguish positive class samples. Equation (8) listed below calculates precision.
(8)$$\mathrm{precision}=\frac{TP}{TP+FP}$$

Recall represents the ratio of the number of samples that the model correctly identified as positive classes to the total number of positive samples. Equation (9) listed below calculates recall.
(9)$$\mathrm{recall}=\frac{TP}{TP+FN}\;$$

The F1 score is a measure of the classification problem. Precision and recall are contradictory metrics. When the value of precision is high, the value of recall is often low. Therefore, we need to consider both metrics together to evaluate the effect of the model, and the F1 score is the harmonized average of precision and recall. Equation (10) calculates the F1 score, where *P* represents precision and *R* represents recall.
(10)$$F1=\frac{2PR}{P+R}=\frac{2TP}{2TP+FP+FN}$$

Accuracy is the most commonly used classification performance metric. It can be used to express the accuracy of the model, that is, the ratio of the number of samples that the model properly identified as positive classes to the total number of samples. Accuracy can be calculated using Equation (11).
(11)$$\mathrm{accuracy}=\frac{TP+TN}{\begin{array}{c} TP+FN+FP+TN \end{array}}$$

IoU (Intersection over Union) is often used to measure the degree of overlap between the predicted box and the ground truth box to evaluate the accuracy of the target detection algorithm. In this paper, IoU is used to measure the ratio of intersection to union set between the bounding boxes of predicted birds and the bounding boxes of real birds. Equation (12) below calculates the IoU.
(12)$$\mathrm{IoU}=\frac{A\cap B}{\begin{array}{c} A\cup B \end{array}}$$

AP (Average Precision) refers to the average precision, that is, the precision of each species is averaged for multi-species prediction, which can measure the effect of the model on each species. We use the integral to calculate the area enclosed by the P-R curve and the coordinate axis to find the AP. The P-R curve is plotted according to the precision and recall of each species. The AP can be calculated using the following Equation (13).
(13)$$\mathrm{AP}=\int_{0}^{1}p\left(t\right)dt$$

MAP (Mean Average Precision) refers to the average of the AP across all species. The mAP is usually used as the final indicator to assess the performance of the metric, measuring the effectiveness of the model on all species. The mAP is calculated by Equation (14), where *S* is the total number of species and the denominator is the sum of the AP under all species.
(14)$$\mathrm{mAP}=\frac{\sum_{j=1}^{S}AP\left(j\right)}{S}$$

FPS (Frames Per Second) is the frame rate per second. Another important metric for the target detection algorithm is the speed and it measures the number of images that the network can process per second. The higher the FPS, the better the timeliness.

### 3.4. Experimental Results

#### 3.4.1. Comparison Experiments of the Most Advanced Methods for Object Detection under Different Labeling Methods

The method for bird monitoring proposed in this paper needs to have a good-performance object detection network for bird species detection. At the same time, it is necessary to use the multi-target tracking network to assign a unique ID to each object to assist the classification count. The multi-target tracking adopts the tracking-by-detection method, and its effect depends on the effectiveness of object detection. Therefore, we need to investigate a high-precision and high-performance object detection network.

We chose the mainstream object detection networks in recent years for comparison (Faster-RCNN [57], EfficientDet [58], CenterNet [59], SSD [60], YOLOv4, YOLOv5, YOLOv7, YOLOv8). Considering the occlusion between birds and the problem of image noise, we not only use the dataset labeled on the bird’s body, but also use the dataset labeled only on the bird’s head to train and test the object detection network, and then combine the experimental results to compare their F1 score, mAP@0.5, FPS and other evaluation indicators. Figure 12 shows the variation of mAP@0.5 for each target detection network during the training period. Table 6 and Table 7 show the comparative experimental results of the object detection networks trained using the datasets of the two labeling methods, respectively.

The experimental results show that the YOLOv7 network trained with a dataset labeling the entire body of the bird is the most effective method. Therefore, we choose YOLOv7 as the research object of this paper and use the dataset annotating the whole body of the bird for subsequent experiments.

#### 3.4.2. Ablation Experiment on Data Enhancement

We tried some training tricks based on the original YOLOv7 model, divided the experimental groups using combining or splitting tricks, and obtained the experimental results shown in Table 8 below through training and testing. The results of this ablation experiment show that when we use HSV, Mosaic, and mixup data augmentation simultaneously, the method of group 12 has the best experimental effect, the evaluation indexes, such as mAP and F1 score, improve the most, and the FPS reduces but still meets the real-time requirement.

#### 3.4.3. Ablation Experiment of Introducing a Series of Improved Strategies for YOLOv7

We reduce information dispersion and amplify the global interaction representation in this paper by adding three GAM modules to the head side of YOLOv7 and replacing the loss function with Alpha-IoU loss to achieve more accurate bounding box regression and object detection. We conducted ablation experiments to validate the method’s effectiveness, and the results show that our method improved the performance of the original YOLOv7 network. Figure 13 shows the variation of mAP@0.5 for each object detection network during the training period. Table 9 shows the experimental results of the ablation experiments.

For commonly used deep learning networks (such as CNNs) are generally considered to be black boxes and not very interpretable. To help better understand and explain the principle and decision-making process of our improvement work, we introduced Grad-CAM [61] in the ablation experiment. The class activation mapping image is generated, which can help us analyze the network’s attention area for a certain species. We can, in turn, analyze whether the network learns the correct features or information through the area of network attention. The heat map drawn by Grad-CAM is shown in Figure 14. From the graph, it can be seen that the improved method proposed in this paper can better mine the structural characteristics of our birds, and it is less affected by image noise. This method is better and more reasonable.

#### 3.4.4. Manual Verification of Algorithm Effectiveness

The monitoring method proposed in this paper uses YOLOv7 and DeepSORT networks to detect and track birds and combines the self-designed counting method to classify and count birds. The overall process is shown in Figure 15, and Figure 16 shows the interface design for the bird monitoring system that we proposed based on actual monitoring results.

We manually counted the species and numbers of birds in the counting area of the video frame to simulate a real-world situation and compared the model’s results of counting birds to those results. The interval of time is neither excessively long nor excessively short for a better performance compared to the performance of each period. The interval of time chosen by this experiment’s counting results is 15 s. Table 10 presents the experimental outcomes.

The results of the verification experiment indicate that the used model’s monitoring of birds is consistent with the actual values. Our method is efficient and feasible, which can assist personnel in understanding the distribution of bird populations and formulate targeted bird protection strategies, thus significantly improving the efficiency of bird protection.

## 4. Discussion

In this paper, we have developed an integrated framework based on computer vision technology for real-time automatic classification and the counting of birds. By using automated monitoring techniques, such as the computer vision monitoring method proposed in this study or sound sensors, it is possible to collect information on birds in remote areas over extended periods of time, thereby increasing the likelihood of discovering rare species [62,63]. Researchers can combine the location information of monitoring sites to collect information on the distribution of bird populations and migration routes in order to develop more effective bird conservation plans. This method plays a certain role in the field of bird conservation, improving the efficiency and accuracy of bird monitoring and making bird conservation more effective [64].

The traditional method of using sample points requires human observers to observe at various sample points within a certain space, which has obvious limitations. For example, in studies of site occupancy or habitat preference [65,66], these limitations are particularly evident in species with low detectability. In such cases, increasing the duration of observations (such as by using automated monitoring techniques) may improve detectability [67,68,69]. Additionally, birds typically inhabit unique and often remote environments, such as dense forests and high-altitude mountains [70], and their small size and large numbers make close observation challenging for researchers. Due to birds’ sensitivity to human disturbance, even slight disturbances can cause significant behavioral reactions and potentially have negative impacts on their survival [71]. In this study, we used deep learning-based techniques for bird species classification and tracking and developed an automated method for classification and counting that can help address the above problems.

Budka et al., used a scientometric analysis to examine publication and research trends [20] and found that in recent years, most publications related to bird monitoring or classification are related to “deep learning.” This indicates that applying deep learning to bird monitoring is a rapidly developing research topic, although the overall research quantity is still limited, which confirms the necessity of using deep learning for bird monitoring. In this paper, we experimented with eight models: Faster-RCNN, EfficientDet, CenterNet, SSD, YOLOv4, YOLOv5, YOLOv7, and YOLOv8 to investigate suitable computer vision methods. Among the eight models, YOLOv7 achieved the best performance, and we further optimized this algorithm. Currently, we have not found scholars who use similar methods for bird monitoring, so we can only compare our model with articles that use different methods but similar tasks. Our proposed model achieved an average precision of over 71% [72], 88.1% [73], and 88.7% [74], which validates its effectiveness in identifying bird categories compared to these methods. Combined with our tracking and counting method, this opens up new perspectives for bird monitoring.

However, our method still has certain limitations, such as the need to improve recognition accuracy in complex backgrounds and lighting conditions. Future research could further explore how to optimize deep learning models to address these challenges, as well as integrate other auxiliary technologies (such as drones, satellite remote sensing, and bioacoustics) [13,14,15,16,17,18,19,20,21,22,23] into the bird monitoring system. Additionally, we could add more negative samples during the training phase and use image datasets generated by GAN [75] for data augmentation to improve monitoring effectiveness.

## 5. Conclusions

It is worth mentioning that this study has constructed a bird species detection and tracking dataset. The dataset includes 3737 images of birds and 11,139 images of the whole body of individual birds, with manual data annotation of the whole body and head of each bird. Such a dataset could provide data to support future bird conservation research. This study proposes a bird monitoring method using computer vision tricks, which uses object detection and a multi-object tracking network to detect and track birds by species, and then combines the information from detection and tracking to count birds by species using an area counting method. We also improved the object detection part by taking YOLOv7, the current mainstream object detection network, as the baseline and fused GAM to the head side of YOLOv7 and changed it to Alpha-IoU loss to obtain a more accurate edge regression and object detection. The improvement resulted in a mAP of 95.1%, which is probably the best model in the field to date. Our experiments have shown that our method can be effective in monitoring birds and obtaining their population distribution, which meets the requirements for practical applications.

In the future, we will continue to optimize the method to achieve better results in more scenarios. For example, we could try to train the model using more datasets or use more advanced tricks to analyze the sounds and behavior of birds. We believe that with these improvements our method will better serve the cause of bird conservation and provide more help for ornithological research.

## Figures and Tables

**Figure 1 animals-13-01713-f001:**
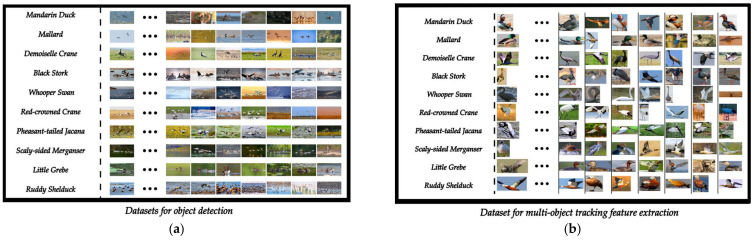
A preview image of the dataset. (**a**) The preview shows the dataset for object detection; (**b**) the preview shows the dataset for the multi-object tracking feature extraction network.

**Figure 2 animals-13-01713-f002:**
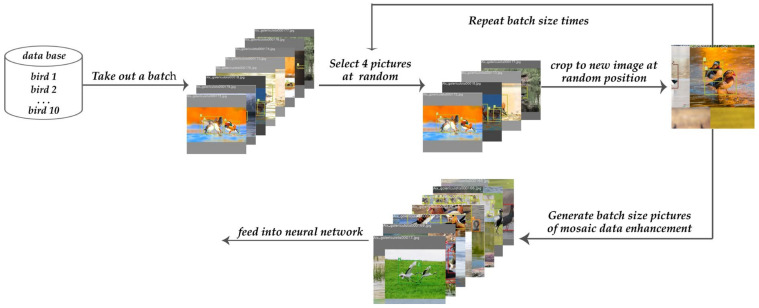
Mosaic data augmentation. First, from the dataset of birds, a batch of image data was randomly extracted. Then, four images were randomly selected, randomly scaled, randomly distributed, and spliced into new images, and the above operations were repeated for batch size times. Finally, the neural network was trained using the Mosaic data augmentation data.

**Figure 3 animals-13-01713-f003:**
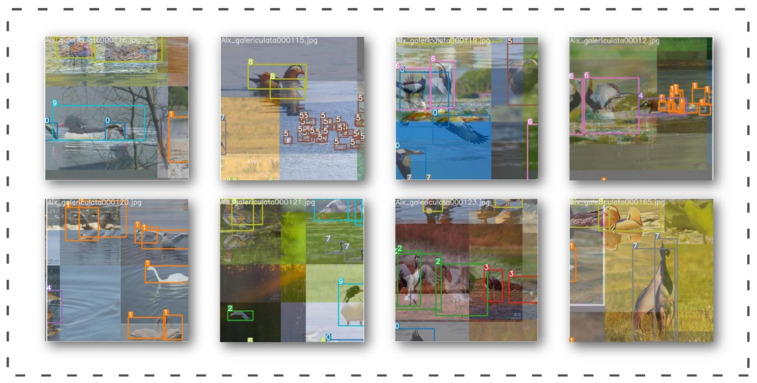
Images after the mixup augmentation processing.

**Figure 4 animals-13-01713-f004:**
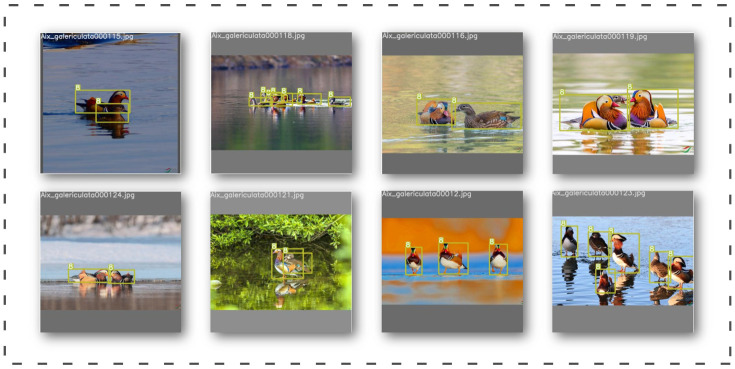
Images after HSV augmentation processing.

**Figure 5 animals-13-01713-f005:**
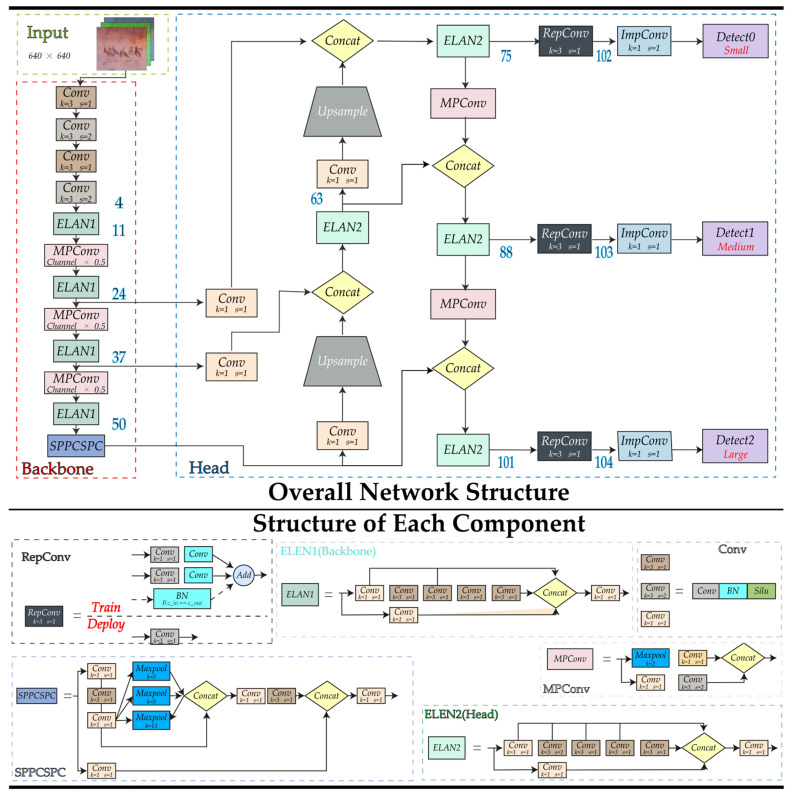
The network architecture diagram of YOLOv7.

**Figure 6 animals-13-01713-f006:**
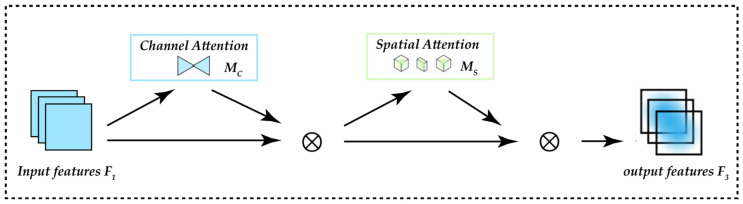
The overview of GAM.

**Figure 7 animals-13-01713-f007:**
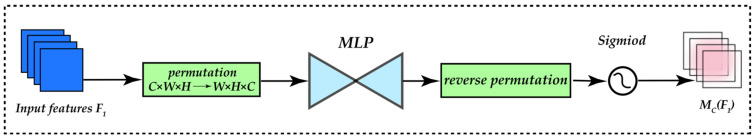
Channel attention submodule.

**Figure 8 animals-13-01713-f008:**
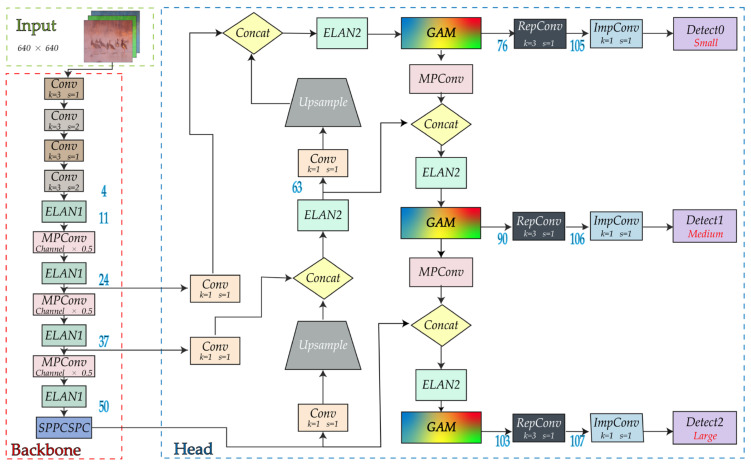
The network structure after adding GAM in YOLOv7.

**Figure 9 animals-13-01713-f009:**
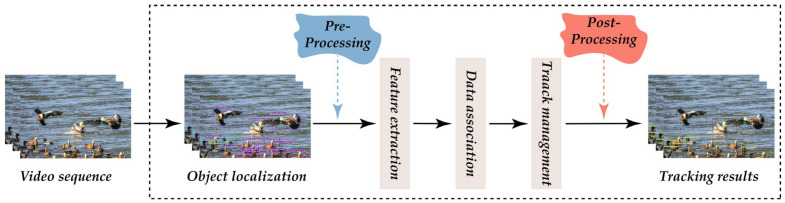
The flowchart of bird tracking.

**Figure 10 animals-13-01713-f010:**
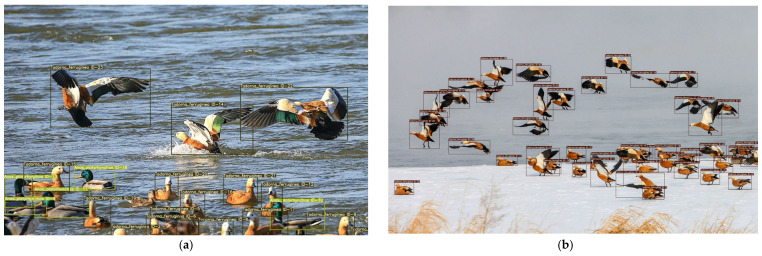
(**a**,**b**) The results of bird tracking.

**Figure 11 animals-13-01713-f011:**
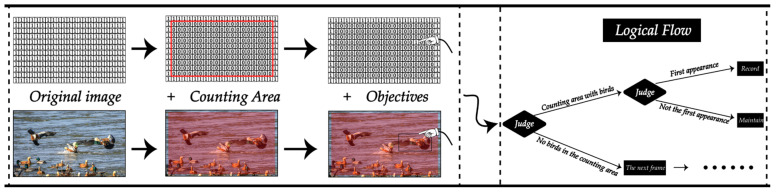
The logic diagram for bird counting.

**Figure 12 animals-13-01713-f012:**
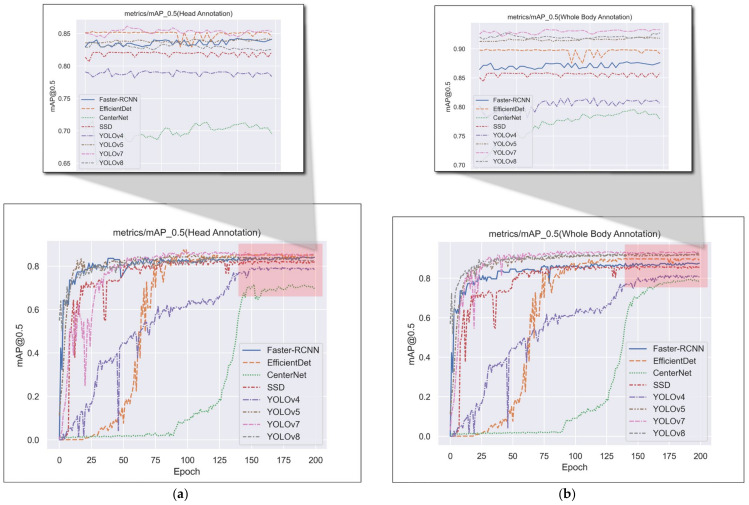
The variation in mAP@0.5 during training. (**a**) Training using the dataset labeled only with the head of the bird; (**b**) training using the dataset labeled with the body of the bird.

**Figure 13 animals-13-01713-f013:**
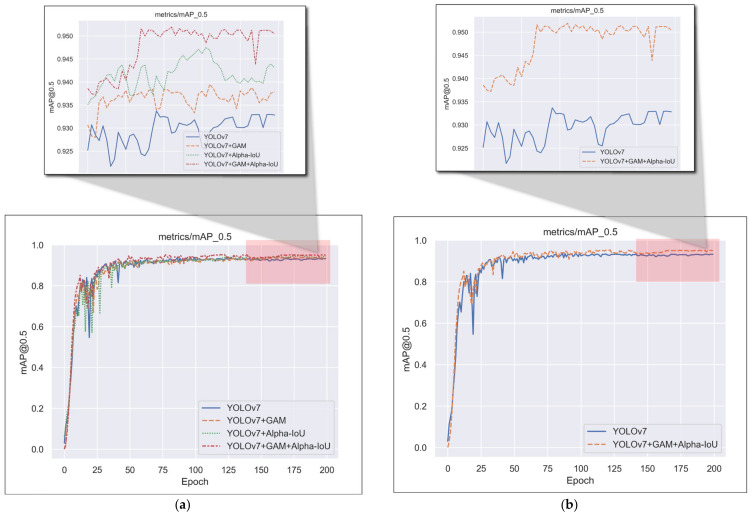
The variation in mAP@0.5 during training. (**a**) Ablation experiments with improved strategies; (**b**) a comparison of the original model with our final improved model.

**Figure 14 animals-13-01713-f014:**
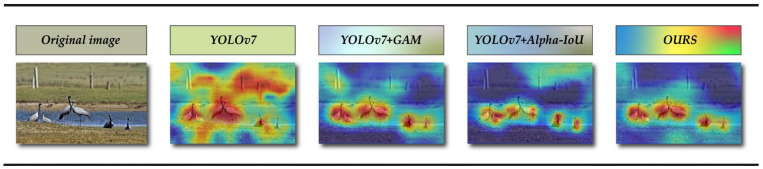
The heat map of various models in the ablation experiment.

**Figure 15 animals-13-01713-f015:**
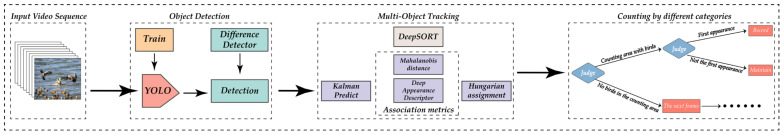
Flowchart of the processing of the method.

**Figure 16 animals-13-01713-f016:**
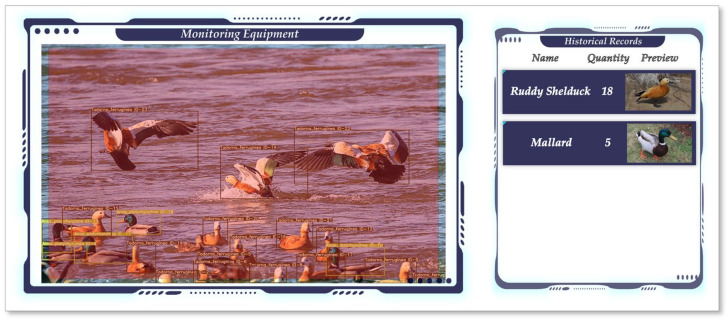
The interface design for the monitoring system.

**Table 1 animals-13-01713-t001:** Partitioning of the datasets for object detection.

Annotation Method	Name	Proportion	Number of Pictures	Number of Birds
Whole Body Annotation	training set	85%	3176	11,322
	validation set	10%	373	1543
	test set	5%	188	863
Head Annotation	training set	85%	2782	10,085
	validation set	10%	327	1217
	test set	5%	164	681
Total	Whole Body Annotation	100%	3737	13,728
	Head Annotation	100%	3273	11,983

Head annotation means that only the bird’s head is annotated, whereas whole body annotation means that the bird’s entire body, including the head, is annotated.

**Table 2 animals-13-01713-t002:** Partitioning of the datasets for multi-target tracking feature extraction networks.

Partition Name	Proportion	Number of Pictures
training set	85%	9468
validation set	10%	1114
test set	5%	557
Total	100%	11,139

**Table 3 animals-13-01713-t003:** Software and hardware experimental equipment.

Name	Type/Version
Operating system	Ubuntu 20.04
Python version	Python 3.8
Versions of the library	Torch1.9.0 + cu111
Integrated Development Environment	Pycharm 2021.3.3
Central Processing Unit	AMD EPYC 7543 32-Core Processor
Graphics Processing Unit	A40(48 GB) × 2

**Table 4 animals-13-01713-t004:** Parameter configuration for training neural networks.

Parameter	Value	Parameter	Value
Initial Learning Rate	0.01	Weight Decay	0.0005
Momentum	0.937	Batch Size	32
Image Size	640 × 640	Epochs	200

**Table 5 animals-13-01713-t005:** Parameter definitions.

Confusion Matrix		Predicted Results	
		Positive	Negative
Real Results	True	$TP^{1}$	$FN^{2}$
	False	$FP^{3}$	$TN^{4}$

$TP^{1}$ (True Positive): It predicts positive classes as positive classes. $FN^{2}$ (False Negative): It predicts positive classes as negative classes. $FP^{3}$ (False Positive): It predicts negative classes as positive classes. $TN^{4}$ (True Negative): It predicts negative classes as negative classes.

**Table 6 animals-13-01713-t006:** A comparison of different object detection algorithms (using the dataset annotated with only the head).

Model	Class	Precision	Recall	F1 Score	mAP@0.5	FPS
Faster-RCNN	All	0.793	0.854	0.82	0.841	25
	Ruddy Shelduck	0.697	0.874	0.78	0.855	
	Whooper Swan	0.686	0.802	0.74	0.775	
	Red-crowned Crane	0.639	0.860	0.73	0.825	
	Black Stork	0.806	0.847	0.83	0.824	
	Little Grebe	0.856	0.897	0.88	0.873	
	Mallard	0.879	0.855	0.87	0.861	
	Pheasant-tailed Jacana	0.868	0.869	0.87	0.859	
	Demoiselle Crane	0.921	0.818	0.87	0.854	
	Mandarin Duck	0.857	0.865	0.86	0.859	
	Scaly-sided Merganser	0.725	0.852	0.78	0.824	
EfficientDet	All	0.873	0.832	0.85	0.851	12
	Ruddy Shelduck	0.887	0.900	0.89	0.895	
	Whooper Swan	0.820	0.745	0.78	0.807	
	Red-crowned Crane	0.872	0.737	0.80	0.821	
	Black Stork	0.838	0.821	0.83	0.834	
	Little Grebe	0.913	0.892	0.90	0.899	
	Mallard	0.812	0.869	0.84	0.829	
	Pheasant-tailed Jacana	0.897	0.857	0.88	0.871	
	Demoiselle Crane	0.930	0.741	0.82	0.807	
	Mandarin Duck	0.911	0.883	0.90	0.891	
	Scaly-sided Merganser	0.854	0.874	0.86	0.861	
CenterNet	All	0.828	0.611	0.70	0.712	59
	Ruddy Shelduck	1.000	0.749	0.86	0.892	
	Whooper Swan	0.914	0.750	0.82	0.883	
	Red-crowned Crane	0.956	0.733	0.83	0.840	
	Black Stork	0.802	0.630	0.71	0.799	
	Little Grebe	0.644	0.793	0.71	0.636	
	Mallard	0.892	0.600	0.72	0.793	
	Pheasant-tailed Jacana	0.802	0.761	0.78	0.792	
	Demoiselle Crane	0.703	0.612	0.65	0.694	
	Mandarin Duck	0.802	0.393	0.53	0.611	
	Scaly-sided Merganser	0.762	0.093	0.17	0.181	
SSD	All	0.861	0.768	0.81	0.821	63
	Ruddy Shelduck	0.790	0.810	0.80	0.809	
	Whooper Swan	0.759	0.623	0.68	0.673	
	Red-crowned Crane	0.854	0.707	0.77	0.811	
	Black Stork	0.890	0.790	0.84	0.840	
	Little Grebe	0.934	0.885	0.91	0.892	
	Mallard	0.867	0.844	0.86	0.866	
	Pheasant-tailed Jacana	0.926	0.892	0.91	0.917	
	Demoiselle Crane	0.843	0.586	0.69	0.725	
	Mandarin Duck	0.859	0.722	0.78	0.796	
	Scaly-sided Merganser	0.893	0.821	0.86	0.878	
YOLOv4	All	0.907	0.679	0.76	0.790	40
	Ruddy Shelduck	0.879	0.888	0.88	0.889	
	Whooper Swan	0.839	0.702	0.76	0.808	
	Red-crowned Crane	0.777	0.664	0.72	0.767	
	Black Stork	0.891	0.765	0.82	0.849	
	Little Grebe	0.962	0.839	0.90	0.933	
	Mallard	0.894	0.696	0.78	0.790	
	Pheasant-tailed Jacana	0.985	0.767	0.86	0.900	
	Demoiselle Crane	0.951	0.656	0.78	0.838	
	Mandarin Duck	0.906	0.556	0.69	0.801	
	Scaly-sided Merganser	0.985	0.254	0.40	0.329	
YOLOv5	All	0.923	0.847	0.88	0.841	88
	Ruddy Shelduck	0.818	0.877	0.85	0.811	
	Whooper Swan	0.901	0.578	0.70	0.734	
	Red-crowned Crane	0.942	0.770	0.85	0.675	
	Black Stork	0.961	0.936	0.95	0.914	
	Little Grebe	0.855	0.946	0.90	0.876	
	Mallard	0.940	0.923	0.93	0.918	
	Pheasant-tailed Jacana	0.960	0.950	0.95	0.862	
	Demoiselle Crane	0.971	0.796	0.87	0.830	
	Mandarin Duck	0.917	0.924	0.92	0.914	
	Scaly-sided Merganser	0.967	0.771	0.86	0.878	
YOLOv7	All	0.850	0.836	0.84	0.862	81
	Ruddy Shelduck	0.911	0.668	0.77	0.800	
	Whooper Swan	0.648	0.701	0.67	0.705	
	Red-crowned Crane	0.851	0.846	0.85	0.876	
	Black Stork	0.652	0.759	0.70	0.726	
	Little Grebe	0.968	0.902	0.93	0.968	
	Mallard	0.841	0.900	0.87	0.915	
	Pheasant-tailed Jacana	0.749	0.909	0.82	0.773	
	Demoiselle Crane	0.959	0.784	0.86	0.903	
	Mandarin Duck	0.960	0.958	0.96	0.989	
	Scaly-sided Merganser	0.962	0.934	0.95	0.966	
YOLOv8	All	0.846	0.800	0.82	0.835	91
	Ruddy Shelduck	0.852	0.569	0.68	0.713	
	Whooper Swan	0.646	0.603	0.62	0.573	
	Red-crowned Crane	0.790	0.815	0.80	0.834	
	Black Stork	0.763	0.774	0.77	0.787	
	Little Grebe	0.954	0.961	0.96	0.970	
	Mallard	0.902	0.864	0.88	0.896	
	Pheasant-tailed Jacana	0.749	0.864	0.80	0.791	
	Demoiselle Crane	0.910	0.768	0.83	0.877	
	Mandarin Duck	0.961	0.881	0.92	0.938	
	Scaly-sided Merganser	0.932	0.898	0.91	0.970	

FPS stands for Frames Per Second and mAP@0.5 is an abbreviation for Mean Average Precision when the Intersection over Union (IoU) is set to 0.5.

**Table 7 animals-13-01713-t007:** A comparison of different object detection algorithms (using the dataset annotated with the whole body).

Model	Class	Precision	Recall	F1 Score	mAP@0.5	FPS
Faster-RCNN	All	0.831	0.892	0.86	0.879	26
	Ruddy Shelduck	0.735	0.912	0.81	0.893	
	Whooper Swan	0.724	0.840	0.78	0.813	
	Red-crowned Crane	0.677	0.898	0.77	0.863	
	Black Stork	0.844	0.885	0.86	0.862	
	Little Grebe	0.894	0.935	0.91	0.911	
	Mallard	0.917	0.893	0.91	0.899	
	Pheasant-tailed Jacana	0.906	0.907	0.91	0.897	
	Demoiselle Crane	0.959	0.856	0.90	0.892	
	Mandarin Duck	0.895	0.903	0.90	0.897	
	Scaly-sided Merganser	0.763	0.890	0.82	0.862	
EfficientDet	All	0.915	0.874	0.89	0.898	14
	Ruddy Shelduck	0.932	0.955	0.94	0.945	
	Whooper Swan	0.860	0.785	0.82	0.857	
	Red-crowned Crane	0.912	0.777	0.84	0.867	
	Black Stork	0.893	0.866	0.88	0.884	
	Little Grebe	0.943	0.937	0.94	0.945	
	Mallard	0.867	0.899	0.88	0.875	
	Pheasant-tailed Jacana	0.927	0.912	0.92	0.917	
	Demoiselle Crane	0.985	0.771	0.86	0.853	
	Mandarin Duck	0.941	0.938	0.94	0.937	
	Scaly-sided Merganser	0.894	0.904	0.90	0.897	
CenterNet	All	0.968	0.659	0.74	0.796	58
	Ruddy Shelduck	1.000	0.977	0.99	0.998	
	Whooper Swan	0.984	0.750	0.85	0.783	
	Red-crowned Crane	0.973	0.750	0.85	0.840	
	Black Stork	1.000	0.851	0.92	0.881	
	Little Grebe	0.842	0.800	0.82	0.936	
	Mallard	0.909	0.517	0.66	0.740	
	Pheasant-tailed Jacana	1.000	0.778	0.88	0.906	
	Demoiselle Crane	0.971	0.810	0.88	0.862	
	Mandarin Duck	1.000	0.333	0.50	0.785	
	Scaly-sided Merganser	1.000	0.023	0.04	0.226	
SSD	All	0.901	0.796	0.84	0.858	60
	Ruddy Shelduck	0.838	0.838	0.84	0.857	
	Whooper Swan	0.784	0.661	0.72	0.696	
	Red-crowned Crane	0.892	0.745	0.81	0.849	
	Black Stork	0.909	0.805	0.85	0.878	
	Little Grebe	0.976	0.900	0.94	0.930	
	Mallard	0.882	0.882	0.88	0.904	
	Pheasant-tailed Jacana	0.968	0.930	0.95	0.955	
	Demoiselle Crane	0.910	0.601	0.72	0.763	
	Mandarin Duck	0.941	0.737	0.83	0.834	
	Scaly-sided Merganser	0.912	0.859	0.88	0.916	
YOLOv4	All	0.924	0.683	0.77	0.811	37
	Ruddy Shelduck	0.848	0.907	0.88	0.944	
	Whooper Swan	0.877	0.713	0.79	0.846	
	Red-crowned Crane	0.769	0.625	0.69	0.782	
	Black Stork	0.929	0.776	0.85	0.887	
	Little Grebe	1.000	0.850	0.92	0.948	
	Mallard	0.932	0.707	0.80	0.805	
	Pheasant-tailed Jacana	1.000	0.778	0.88	0.892	
	Demoiselle Crane	0.966	0.667	0.79	0.853	
	Mandarin Duck	0.921	0.556	0.69	0.816	
	Scaly-sided Merganser	1.000	0.250	0.40	0.333	
YOLOv5	All	0.865	0.805	0.83	0.920	93
	Ruddy Shelduck	0.808	0.824	0.82	0.911	
	Whooper Swan	0.897	0.678	0.77	0.734	
	Red-crowned Crane	0.875	0.477	0.62	0.875	
	Black Stork	0.911	0.874	0.89	0.984	
	Little Grebe	0.823	0.894	0.86	0.960	
	Mallard	0.914	0.909	0.91	0.978	
	Pheasant-tailed Jacana	0.769	0.912	0.83	0.992	
	Demoiselle Crane	0.873	0.774	0.82	0.910	
	Mandarin Duck	0.828	0.912	0.87	0.964	
	Scaly-sided Merganser	0.954	0.791	0.86	0.889	
YOLOv7	All	0.942	0.870	0.90	0.932	111
	Ruddy Shelduck	0.766	0.896	0.83	0.921	
	Whooper Swan	0.929	0.601	0.73	0.760	
	Red-crowned Crane	0.975	0.809	0.88	0.915	
	Black Stork	0.984	0.947	0.97	0.980	
	Little Grebe	0.900	0.973	0.94	0.974	
	Mallard	0.951	0.939	0.95	0.978	
	Pheasant-tailed Jacana	1.000	0.987	0.99	0.996	
	Demoiselle Crane	0.983	0.830	0.90	0.915	
	Mandarin Duck	0.967	0.930	0.95	0.960	
	Scaly-sided Merganser	0.966	0.791	0.87	0.921	
YOLOv8	All	0.925	0.864	0.89	0.927	97
	Ruddy Shelduck	0.846	0.900	0.87	0.929	
	Whooper Swan	0.892	0.597	0.72	0.759	
	Red-crowned Crane	0.971	0.799	0.88	0.899	
	Black Stork	0.953	0.947	0.95	0.976	
	Little Grebe	0.852	0.946	0.90	0.970	
	Mallard	0.946	0.937	0.94	0.971	
	Pheasant-tailed Jacana	1.000	0.975	0.99	0.995	
	Demoiselle Crane	0.975	0.859	0.91	0.947	
	Mandarin Duck	0.897	0.930	0.91	0.953	
	Scaly-sided Merganser	0.954	0.755	0.84	0.875	

FPS stands for Frames Per Second and mAP@0.5 is an abbreviation for Mean Average Precision when the Intersection over Union (IoU) is set to 0.5.

**Table 8 animals-13-01713-t008:** Each experimental group in YOLOv7′s data-enhanced ablation experiments corresponds to a group of tricks and evaluation metrics. The “✓“ indicates that the trick is not used in this group of experiments and the “🗴“ indicates that it is used in this group of experiments.

Group	HSV	Mosaic	MixUp	FocalLoss	Precision	Recall	F1 Score	mAP@0.5	mAP@0.5:0.95	FPS
1	🗴	🗴	🗴	🗴	0.939	0.870	0.90	0.932	0.807	111
2	✓	🗴	🗴	🗴	0.914	0.885	0.90	0.930	0.801	92
3	🗴	✓	🗴	🗴	0.933	0.876	0.90	0.931	0.798	89
4	🗴	🗴	✓	🗴	0.929	0.878	0.90	0.929	0.798	91
5	🗴	🗴	🗴	✓	0.925	0.872	0.90	0.924	0.782	82
6	✓	✓	🗴	🗴	0.924	0.885	0.90	0.930	0.801	84
7	✓	🗴	✓	🗴	0.935	0.880	0.91	0.927	0.790	79
8	✓	🗴	🗴	✓	0.913	0.877	0.89	0.929	0.801	83
9	🗴	✓	✓	🗴	0.940	0.881	0.91	0.932	0.807	86
10	🗴	✓	🗴	✓	0.916	0.884	0.90	0.929	0.777	77
11	🗴	🗴	✓	✓	0.930	0.874	0.90	0.929	0.797	82
12	✓	✓	✓	🗴	0.942	0.888	0.91	0.933	0.809	85
13	✓	✓	🗴	✓	0.932	0.876	0.90	0.927	0.783	81
14	✓	🗴	✓	✓	0.933	0.879	0.91	0.927	0.789	81
15	🗴	✓	✓	✓	0.945	0.879	0.91	0.931	0.801	80
16	✓	✓	✓	✓	0.932	0.875	0.90	0.927	0.788	84

FPS stands for Frames Per Second and mAP@0.5 is an abbreviation for Mean Average Precision when the Intersection over Union (IoU) is set to 0.5.

**Table 9 animals-13-01713-t009:** Ablation experiments with improved algorithms. (For the following experiments, mixup, Mosaic, and HSV data augmentation methods are used by default.)

Model	Class	Precision	Recall	F1 Score	mAP@0.5	mAP@0.5:0.95	FPS
YOLOv7	All	0.942	0.888	0.91	0.933	0.809	85
	Ruddy Shelduck	0.766	0.912	0.83	0.921	0.804	
	Whooper Swan	0.931	0.669	0.78	0.770	0.536	
	Red-crowned Crane	0.975	0.819	0.89	0.915	0.757	
	Black Stork	0.984	0.956	0.97	0.980	0.864	
	Little Grebe	0.900	0.979	0.94	0.974	0.931	
	Mallard	0.951	0.939	0.94	0.978	0.878	
	Pheasant-tailed Jacana	1.000	0.957	0.98	0.996	0.912	
	Demoiselle Crane	0.983	0.883	0.93	0.915	0.760	
	Mandarin Duck	0.967	0.934	0.95	0.960	0.852	
	Scaly-sided Merganser	0.966	0.833	0.89	0.921	0.793	
YOLOv7 + GAM	All	0.929	0.883	0.91	0.938	0.803	101
	Ruddy Shelduck	0.750	0.890	0.81	0.917	0.792	
	Whooper Swan	0.920	0.649	0.76	0.780	0.538	
	Red-crowned Crane	0.972	0.834	0.90	0.922	0.754	
	Black Stork	0.956	0.962	0.96	0.979	0.849	
	Little Grebe	0.818	0.973	0.89	0.982	0.923	
	Mallard	0.947	0.961	0.95	0.977	0.884	
	Pheasant-tailed Jacana	1.000	0.987	0.99	0.996	0.889	
	Demoiselle Crane	0.983	0.834	0.90	0.921	0.752	
	Mandarin Duck	0.967	0.939	0.95	0.971	0.851	
	Scaly-sided Merganser	0.978	0.799	0.88	0.931	0.797	
YOLOv7 + Alpha-IoU	All	0.945	0.887	0.92	0.947	0.809	92
	Ruddy Shelduck	0.873	0.908	0.89	0.944	0.816	
	Whooper Swan	0.914	0.684	0.78	0.811	0.549	
	Red-crowned Crane	0.972	0.825	0.89	0.935	0.747	
	Black Stork	0.952	0.962	0.96	0.980	0.867	
	Little Grebe	0.885	0.973	0.93	0.989	0.906	
	Mallard	0.946	0.947	0.95	0.979	0.879	
	Pheasant-tailed Jacana	1.000	0.987	0.99	0.995	0.901	
	Demoiselle Crane	0.972	0.828	0.89	0.920	0.778	
	Mandarin Duck	0.968	0.943	0.96	0.978	0.854	
	Scaly-sided Merganser	0.971	0.817	0.89	0.940	0.793	
YOLOv7 + GAM + Alpha-IoU (YOLOv7^Birds^)	All	0.945	0.898	0.92	0.951	0.815	82
	Ruddy Shelduck	0.892	0.904	0.90	0.944	0.803	
	Whooper Swan	0.922	0.730	0.81	0.825	0.573	
	Red-crowned Crane	0.984	0.858	0.92	0.935	0.752	
	Black Stork	0.947	0.962	0.95	0.981	0.857	
	Little Grebe	0.817	0.973	0.89	0.990	0.919	
	Mallard	0.956	0.946	0.95	0.983	0.880	
	Pheasant-tailed Jacana	1.000	0.987	0.99	0.996	0.911	
	Demoiselle Crane	0.977	0.871	0.92	0.932	0.807	
	Mandarin Duck	0.975	0.943	0.96	0.985	0.847	
	Scaly-sided Merganser	0.977	0.808	0.88	0.937	0.797	

FPS stands for Frames Per Second and mAP@0.5 is an abbreviation for Mean Average Precision when the Intersection over Union (IoU) is set to 0.5.

**Table 10 animals-13-01713-t010:** Comparison between quantity in reality and quantity calculated by the model.

Interval of Time		0–15 s	16–30 s	31–45 s	46–60 s
Results of manual count (“true count”)	Quantity	36	44	65	72
	Counting Accuracy (%)	100%	100%	100%	100%
Results of our algorithm	Quantity	36	44	65	72
	Counting Accuracy (%)	100%	100%	100%	100%

## Data Availability

The data are available online at https://www.kaggle.com/datasets/dieselcx/birds-chenxian, accessed on 12 May 2023.

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
