# Peer review of "An Efficient Method for Monitoring Birds Based on Object Detection and Multi-Object Tracking Networks"

_animals, 2023, doi:10.3390/ani13101713_

Round 1

Reviewer 1 Report

The problem of accurate counting of birds in flocks, their species identification in aggregations is important for field ornithology. Regular monitoring, both in protected areas and in human-made and operated facilities that may have an impact on birds, is impossible without accurate identification and accounting. This traditional method, indeed, requires a lot of time and labor, highly qualified observers, which allows minimizing errors in counting and identifying birds. It is still difficult to imagine that bird monitoring can be carried out only by technical means without human participation at the field stage. The introduction of image (face) recognition technologies has been used in various fields for a long time. It is probably the time when one can try to introduce bird recognition technologies into practice. The authors of the peer-reviewed manuscript describe the successes and limitations of previous research in this direction. Such design can be useful in certain situations in order to reduce the labor costs associated with the organization and conduct of bird observations by humans in the field. The manuscript under review is devoted to the development of such a method using the example of 10 waterfowl species that can be identified from images. Large birds that can concentrate in open spaces of water bodies are selected for recognition. The authors point out the limitations of their bird recognition method. However, it should be noted that they have made progress, significantly reducing the errors in species identification and quantitative counting of 10 bird species. While the method is far from being implemented in the practical field, however, I believe that the manuscript can be published.

The technique is described in sufficient detail and may be one of the steps in the development of a truly useful ornithological monitoring method.

The manuscript has some editorial and technical comments.

 1. I suggest adding to the keywords: "bird monitoring".

2. The goal of the study (lines 150-152) should be moved to the Introduction section .

3. I did not find an explanation in the text why these 10 species were chosen for recognition: are they large and easy to identify? Or is there some other meaning in this choice?

4. I think that the word “bird category” should be replaced everywhere in the text on “bird species”, since the purpose of the study is to recognize certain species of birds.

5. It is necessary to repeat the title line of long tables on the following pages for ease of perception (tables 6,7,9).

6. Latin species names in biology are usually written in italic font.

7. You need to update the reference [3].

Reviewer 2 Report

The modelling aspect to this work is very impressive, but the conservation aspect needs some work. I have left comments on the PDF throughout the paper. The introduction and discussion need a lot of work to get the writing to the correct level. There are many places throughout that are lacking references, and your references need to be more global.

You need to introduce earlier in the paper that you are working with still oblique images. I was left wondering for a very long time whether you were working with camera traps, hand held, video, drone imagery etc.

My biggest critique on the methods / results section from my understanding is that you've been selective about which results to present. You've presented very high accuracy results for counts <35 birds, but as a reader or person who would potentially like to implement this method I need to know the accuracies when there are more than 35 birds in view, even if these accuracies aren't as high. It is very unrealistic to present a model that only works when theres less than 35 birds, knowing birds often aggregate in hundreds, to thousands of individuals. It is ok to present results that aren't perfect, and in fact, it is very important to present these results as they give a realistic expectation of AI methods for counting.

I also think you need to be careful about making statements like "worlds first labelled wetland bird dataset" -- as this statement is largely untrue. Perhaps it is the first published using these species in this method - but there are many datasets in production around the world working in this field.

Overall I think the modelling and potential outcomes of this work are very impressive and creative so very good work on that!
